# Host-symbiont stress response to lack-of-sulfide in the giant ciliate mutualism

**Salvador Espada-Hinojosa** [ID] *, **Judith Drexel, Julia Kesting, Edwin Kniha**¤a, **Iason Pifeas, Lukas Schuster**¤b, **Jean-Marie Volland** [ID]¤c, **Helena C. Zambalos, Monika Bright**

Department of Functional and Evolutionary Ecology, University of Vienna, Vienna, Austria

¤a Current address: Institute of Specific Prophylaxis and Tropical Medicine, Center for Pathophysiology, Infectiology and Immunology, Medical University of Vienna, Vienna, Austria
¤b Current address: Centre for Geometric Biology, School of Biological Sciences, Monash University, Melbourne, Victoria, Australia
¤c Current address: DOE JGI/LRC Systems, Menlo Park, California, United States of America
* salvador.espada@univie.ac.at

**Data Availability Statement:** All data and code files are available from the Dryad repository (doi:10. 5061/dryad.vt4b8gts8).

## Abstract

The mutualism between the thioautotrophic bacterial ectosymbiont *Candidatus* Thiobius zoothamnicola and the giant ciliate *Zoothamnium niveum* thrives in a variety of shallow-water marine environments with highly fluctuating sulfide emissions. To persist over time, both partners must reproduce and ensure the transmission of symbionts before the sulfide stops, which enables carbon fixation of the symbiont and nourishment of the host. We experimentally investigated the response of this mutualism to depletion of sulfide. We found that colonies released some initially present but also newly produced macrozooids until death, but in fewer numbers than when exposed to sulfide. The symbionts on the colonies proliferated less without sulfide, and became larger and more rod-shaped than symbionts from freshly collected colonies that were exposed to sulfide and oxygen. The symbiotic monolayer was severely disturbed by growth of other microbes and loss of symbionts. We conclude that the response of both partners to the termination of sulfide emission was remarkably quick. The development and the release of swarmers continued until host died and thus this behavior contributed to the continuation of the association.

## Introduction

While aerobic eukaryotes die after prolonged exposure to one of the most dangerous poisons, hydrogen sulfide ([1]; $\Sigma H_2S$ i.e. sum of all forms of dissolved sulfide [2], hereafter called sulfide) most mutualistic associations between protist or animal hosts and thioautotrophic bacterial symbionts depend on the presence of sulfide (see [3]). These symbionts share the need for reduced sulfur species (e.g. exclusively sulfide or additionally thiosulfate) and oxygen or alternative electron acceptors to gain energy for carbon fixation [3, 4]. The hosts provide sulfide and oxygen through uptake and transport to the symbionts or through specific behavior such as swimming in and out of vent fluid in shrimps, contraction/expansion behavior in colonial

**Funding:** Funding came from two Austrian Science Fund projects FWF P24565 B22 and FWF 32197 granted to MB. (https://www.fwf.ac.at). The funders had no role in study design, data collection and analysis, decision to publish, or preparation of the manuscript.

**Competing interests:** The authors have declared that no competing interests exist.

ciliates, or digging with the foot in some bivalves (see [3, 5, 6]). As far as is known, the hosts are also able of detoxify sulfide (see [1]). In return, the symbionts nourish their hosts (see [3]).

Many habitats of these symbioses are relatively short-lived, such as fast-spreading deep-sea hydrothermal vents, whale and wood falls depending on substrate size, and decaying seagrass debris (see [3]). In contrast to geothermally generated hydrogen sulfide as in vents (see [7]), the biological sulfide production by sulfate-reducing bacteria ceases when organic material is depleted (see [8]). Upon changes in chemical conditions, mobile animal hosts e.g. stilbonematine nematodes, gutless oligochaetes, snails and bathymodiolin mussels, can migrate to more suitable habitats (see [3]). However, sessile hosts like siboglinid tubeworms [9] or peritrich ciliates [10–13] do not have this option.

To persist over generations, hosts reproduce primarily by releasing motile larvae into the pelagial. Regardless of their mobility as adults, larvae of bathymodiolin mussels, lucinid clams, and tubeworms spread without their symbionts [14]. In these systems host reproduction and symbiont transmission are decoupled and the uptake of symbionts from a free-living population takes place after the larvae have settled. Experiments with some of these symbioses in oxic, non-sulfidic seawater showed that bathymodiolins and lucinids either lost their symbionts or greatly reduced their density. Nevertheless, hosts were able to survive between one and five months until the end of experiments [15–18]. In contrast, experiments with tubeworms showed that their symbionts escaped after the host died [19]. The larvae of other hosts like vesicomyid and solemyid clams carry their symbionts (see [14]). Whether such hosts with vertically transmitted symbionts react to sulfide deficiency with loss of symbionts has not been investigated. Furthermore, it is not known whether sessile hosts of thioautotrophic symbionts continue to reproduce under stress.

The symbiotic mutualism of the giant colonial ciliate *Zoothamnium niveum* (short *Zoothamnium*) and its thioautotrophic gammaproteobacterial ectosymbiont, originally described as *Cand*. Thiobios zoothamnicoli but due to nomenclature regulations corrected to *Cand*. Thiobius zoothamnicola, ([20], short Thiobius) is a suitable model to study host-symbiont response to environmental stress and disturbance when sulfide ceases. In contrast to slow-growing and reproducing animal hosts, this is a fast-growing and fast reproducing, sessile ciliate [21] which thrives on ephemeral sulfide-emitting surfaces in shallow, tropical to temperate environments such as wood, mangrove peat, decaying seagrass, and whale bones [22].

*Zoothamnium* colonies consist of a stalk and alternating branches on which individual cells grow: feeding cells called microzooids, dividing cells called terminal zooids, and macrozooids, cells responsible for asexual reproduction ([11], Fig 1). The vertical transmission of the ectosymbiont is through macrozooids. These host propagules are released as swarmers into the pelagial for dispersal. Once settled, the swarmer transforms into the terminal zooid and begins to produce the stalk and to divide, producing the terminal zooid of each branch. Nourishing microzooids are produced through division of the terminal zooid on each branch, increasing the length of the branch. Macrozooids develop at the base of the branch. These macrozooids leave the colony as soon as a ciliary band has formed.

The dual partnership involves a single bacterial phylotype covering the host surface in a monolayer with the exception of the lowest, senescent parts of the colonies. There the symbionts become overgrown or replaced by other microbes [12, 23, 24]. The symbiont is rod-shaped on the aboral part of the microzooid and more coccoid-shaped on the oral part [12]. This phenotypic difference was explained by the movement of cilia around the oral ciliature in microzooids, which provides the symbionts with more balanced inorganic compounds for growth compared to all other cilia-free host parts [21].

In nature, host colonies are found at the interface between sulfidic and oxic layers of seawater [25, 26]. The ciliate is capable of rapid contractions of the stalk and branches [11, 12]. With

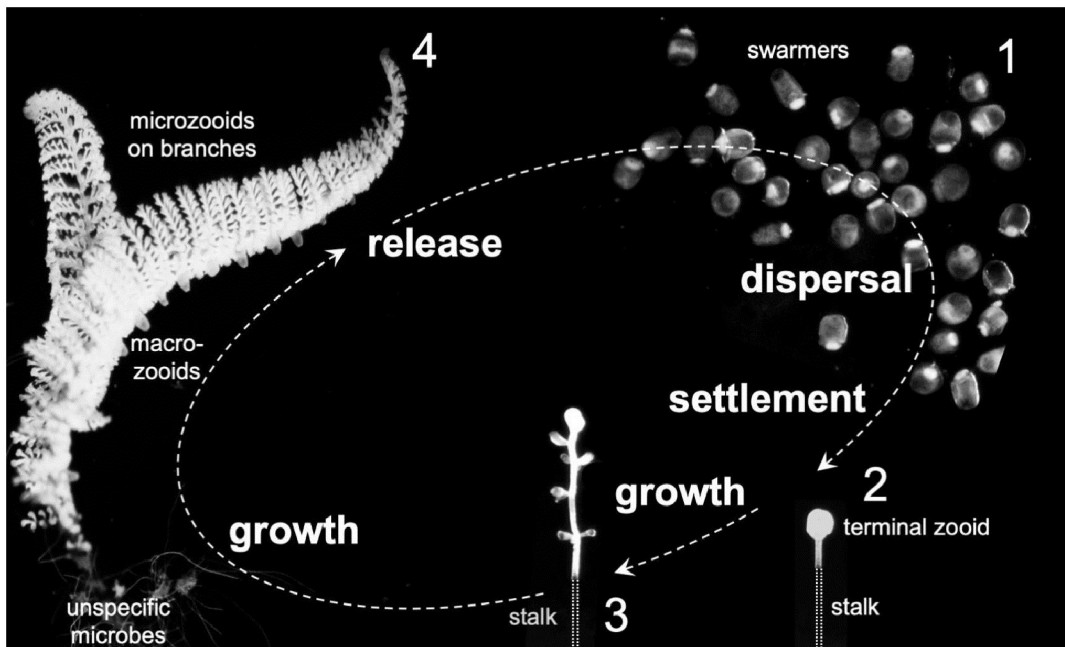

**Fig 1. Life cycle of *Zoothamnium niveum*.** The swarmers are the dispersal stage (1), and look for a sulfide source to settle. Once settled, the swarmer transforms into the terminal zooid at the top of the new colony and grows a stalk. Note that the white part of the stalk is overgrown by the symbiont, but the lower black part is aposymbiotic (2). The terminal zooid divides and produces the terminal zooids for each branch (3). The branch grows by divisions of the terminal zooid on the tip of the branch, creating microzooids and macrozooids, that eventually detach as swarmers (4). Light micrographs not to scale. Note that the lower part of the stalk, lacking white symbionts, is outlined.

this repetitive movement they create a mixture of these chemicals [25, 27–29], which are necessary for host respiration as well as symbiont carbon fixation and translocation of fixed carbon to the host [30]. Carbon fixation in symbionts and direct nutritional transfer to the host has been demonstrated under oxic, sulfide supplemented conditions but ceased under oxic conditions without sulfide [30].

The host colonies are sessile and can only 'escape' sulfide starvation through producing macrozooids and releasing them as swarmers [11, 12]. How quickly colonies die in nature when sulfide stops is unknown. Experiments in the lab showed that all swarmers lost their symbionts in two days and thus became aposymbiotic. Regardless of presence or absence of symbionts, the vast majority only settled in the presence of sulfide [31]. Once they have settled, they can grow aposymbiotically into small colonies with or without sulfide in custom-designed flow-through chambers with a steady flow of oxic seawater. Symbiotic colony growth only happened in symbiotic swarmers exposed to sulfide and oxygen [31]. Fastest colony growth and a life span of 11 days was at low sulfide concentrations [21]. Colonies grew more slowly without sulfide and had a 4.5 days life span [21]. However, we do not know how long large colonies can survive under oxic conditions when sulfide stops and what happens to their symbionts. Furthermore, it is unknown whether the host releases swarmers that carry the symbionts as a response to sulfide starvation and how long these swarmers survive in order to find a suitable habitat for settlement.

Here, we followed the fate of large colonies and their propagules experimentally mimicking the waning of sulfide. We specifically asked how long freshly collected colonies and their swarmers survive under experimental oxic, non-sulfidic conditions (in short sulfide starvation). We also investigated whether the release of macrozooids comes only from those that

were already present on the colony at the beginning of the experiment, or whether the production of new macrozooids and their release as swarmers under sulfide deficiency was continued. For comparison we also performed a sulfidic control experiment. Further we studied the symbiont morphology, their frequency of dividing cells (FDC), as well as their density and coverage on the host surface, and the colonization of other microbes under oxic conditions using scanning electron microscopy (SEM) and fluorescence *in situ* hybridization (FISH). For comparison we used freshly collected colonies from the field. Given the lack of sulfide fueling symbiont carbon fixation [30], we hypothesized that the symbiont division should cease and consequently the monolayer on the host feeding on symbionts [30] should be disrupted. Since the symbiont is known to nourish the host [30] we hypothesized that the presence or absence of sulfide indirectly affects the host's longevity and its macrozooid production and release, which are taken as proxy for reproductive effort.

## Material and methods

### Ethic statement

No specific permissions were required for the listed locations as they are publicly accessible. Furthermore, we confirm that our field studies did not involve endangered or protected species.

### Sampling

*Zoothamnium niveum* colonies were collected from shallow, subtidal submerged woods at two locations in the Northern Adriatic Sea close to Piran, Slovenia: the estuarine canal Sv. Jernej (45˚29'48.6"N, 13˚35'57.0"E) and the mudflat in Strunjan (45˚31'44.0"N, 13˚36'13.2"E). Simultaneously, water samples were taken adjacent to the wood pieces and *in situ* temperature, salinity, and pH were measured using a Multi 340i sensor WTW (S1 Table). Wood pieces were transported in buckets filled with on-site seawater to the laboratory and maintained in flow-through aquaria until colonies were used for the experiments, from immediately after collection up to five days later for the sulfide starvation experiment and 23 days for the sulfidic control experiment. During maintenance about 250 mL of 1 mmol $L^{-1}$ sulfide solution was added to each 50 L liter aquarium daily during which fresh seawater flow was stopped for a few hours. Each colony was cut off the wood with a MicroPoint™ Scissor and cleaned from debris by rinsing it in filtered seawater prior the experimental procedure. All seawater used for this and further procedures was filtered through a 0.2 μm Acrodisc® syringe filter.

### Host response to sulfide starvation compared to sulfidic conditions

We used 60 colonies from each of the two collection sites Sv. Jernej and Strunjan for the sulfide starvation experiment in 2015. For comparison, 60 colonies were sampled for the sulfidic control experiment at Strunjan in 2021. Each colony was placed in a well of a multiwell plate. Each well was filled with 1 mL oxic, filtered seawater (sulfide starvation experiment). For the sulfidic control we added sodium sulfide to the filtered seawater at an average final concentration of 448 μmol $L^{-1}$.

The number of macrozooids present on each colony was counted at the start of the experiment (S1 Fig). Every 12 h, viability of colonies was assessed by their contraction/expansion behavior. Colonies that did not contract when being touched with a dissecting needle were considered dead. All swarmers released from each colony within 12 h time intervals were transferred into individual wells.

Every 12 h about two-thirds of the water from each well was replaced by new filtered oxic seawater (sulfide starvation experiment) or filtered sulfidic seawater (sulfide control experiment). The removed water was pooled for measurements of temperature, salinity, pH, and oxygen concentration (S2 Table). Oxygen and temperature were measured with a PreSenS Flow-through Cell FTC-PSt3. Salinity and pH and were measured with a Multi 340i sensor WTW. Additionally, sulfide concentration was measured photometrically according to Cline [32] in a few randomly chosen wells in the sulfidic control experiment in the newly prepared and in the removed water.

To estimate colony size, the number of branches was counted either after host death or at the end of the experiment (sulfide starvation experiment: n = 85; sulfidic control experiment: n = 60). Swarmers from the sulfide starvation experiment were mounted on glass slides and their body size was estimated using Leica DM2000 light microscope equipped with a Leica DFC295 camera and the image analysis software Gimp (GNU Image Manipulation Program) for Mac 2.8.

For statistical comparisons of the colonies used for the sulfide starvation experiment sampled from two locations, 60 colonies from each location were divided into four batches (A-D), with 15 colonies each. The size of the swarmers was measured according to the timeframe of 12-h-interval observations they were released from the colony and the timeframe they were kept in the water swimming (batch A 0 h, B 24 h, C 48 h, D 72 h; S2 Fig). All time points were considered as the upper bound of the intervals. Statistical analyses were conducted in PAST 3.04 [33] and R [34]. Because Shapiro-Wilk tests showed deviations from normality for all parameters (counted branches taken as estimate of colony size, initial macrozooids per colony, total released swarmers per colony, and swarmer size) the Wilcoxon-Mann-Whitney test for equal medians was used for comparisons of the two locations.

In both, the sulfide starvation and sulfidic control experiments, the mortalities of colonies and swarmers were estimated as the proportion of dead colonies/swarmers to the total number of colonies/swarmers used in the experiment. $LT_{50}$ and their standard error estimates for colonies and swarmers were obtained by curve fitting of a binomial Generalized Linear Model (bGLM) with mortality rate as response and time as predictor, and the use of the R package MASS version 7.3–51.4. Goodness of fit was characterized with the Deviance ($D^2$) = (Null Deviance-Residual Deviance)/(Null Deviance). Ordinary Least Squares (OLS) regression-models were used to depict the correlation between relevant magnitudes, e.g. colony size and released swarmers. The number of macrozooids produced during the experiment ($\Delta_M$) was calculated by subtracting the initial number of macrozooids present on colonies at the beginning of the experiment from the sum of the released swarmers plus the macrozooids remaining on the colony at the end of the experiment (S1 Fig). $\Delta_S$, the number of macrozooids produced and released as swarmers during the experiment, was calculated by the subtraction of the initial number of macrozooids from the number of released swarmers. A positive $\Delta_S$ value indicates additionally produced swarmers during the experimental time frame, whereas a negative $\Delta_S$ value indicates remaining macrozooids on the respective colony that were not released at the end of the experiment. Linear fit slope comparisons between both experiments were performed through analyses of covariance by obtaining the significance of the interaction with R [34].

## Symbiont response to sulfide starvation

In 2012, 2013, and 2014, sets of 15 to 20 freshly collected colonies from Sv. Jernej were put into embryo dishes, each kept completely filled with filtered, oxic seawater and covered with glass plates to avoid evaporation for up to 72 h. At the time points 12, 24, 48, and 72 h viability of

colonies was assessed as described above, and at most 3 live colonies were removed and fixed for SEM or divided into live and dead colonies and fixed for FISH (S1 Table). At each time point water was replaced with filtered seawater (S2 Table).

## Fluorescence *in situ* hybridization (FISH)

Colonies from the 2014 embryo dish experiments were fixed and stored in 100% ethanol at 4˚C for 3 months. Colonies were embedded in LR-White resin and polymerization was performed in absence of oxygen at 41˚C for three days. Semi-thin sections (1 μm) were cut on a Reichert Ultracut S microtome, placed in a drop of 20% acetone on chromium(III)potassium sulfate coated glass slides and were left to dry at 40˚C. A total of 16 sections were placed on one slide with four spots of four sections each. To have a representative area of the colony on each slide, two slides per sample were used.

Hybridization was carried out as described in [24]. On each slide the symbiont-specific oligonucleotide probe ZNS196_mod [30] labeled with Cy3 together with a mix of EUB I, II, III (targeting most bacteria [35, 36]), and Arc 915 (targeting most archaea, [37]), all labeled with Cy5 to distinguish the symbiont from any other microbe, were used on two spots. Negative controls with nonsense probes (NON-EUB) labeled in both colors [38] were run on each slide on two different spots. In brief, applied probes were hybridized at 46˚C for 3 hours in the dark, then rinsed in the washing buffer at 48˚C for 15 min, stained with DAPI, washed with Milli Q, and mounted with Citifluor antifading solution. Sections were observed on a Zeiss Axio Imager M2 epifluorescence microscope and images were taken at 100x magnification with an AxioCam MRm, Zeiss using AxioVision Rel. 4.8. software. Composite pictures of entire colony sections were done with ICE software (Image Composite Editor 2.0, Microsoft).

## Scanning electron microscopy (SEM)

Colonies from the 2012 and 2013 embryo dish experiments were placed in a freezer at -20˚C in 2.5 mL of filtered, oxic seawater for 9.5 min prior to fixation to avoid contraction of the colonies [21]. Before the freezing point was reached, the embryo dish was taken out and 2.5 mL of modified Trump´s fixative (2.5% glutaraldehyde, 2% paraformaldehyde in 0.1 M sodium cacodylate buffer 1100 mOsm, pH 7.2, filtered with a 0.2 μm filter prior fixation) was added (modified from [39]). The samples were immediately rinsed with this solution and stored until further treatment.

After storage in fixative for a few months, colonies were rinsed in 0.1 M sodium cacodylate buffer (1100 mOsm, pH 7.2) three times for 3 min each, dehydrated in acetone and transferred to a mixture of acetone/hexamethyldisilazane (HDMS) (1:1) for 15 min, followed by two baths of pure HDMS for 15 min each. Subsequently, the samples were air dried for 3 h, placed on a stub and sputter coated with gold-palladium using an Agar Sputter Coater Agar 108 for 250 seconds.

For detailed SEM observations on a Philips XL 20 scanning electron microscope (acceleration voltage of 20kV) we used two sets of three colonies kept in oxic seawater for 48 h (the set from 2012 was used for statistical analyses, the other one from 2013 for additional SEM micrographs). We used three colonies freshly collected from the environment in 2012 as control. From each colony, images were taken from 15 microzooids (feeding cells) at a magnification of 2000x. The following symbiont parameters were analyzed for the oral and the aboral part of each microzooid separately: 1) number of symbiont cells in a 70 $\mu m^2$ rectangular frame; all cells crossing the edge of the frame were counted only along one length and width of the frame; cells crossing the whole frame either longitudinally or horizontally were also counted. 2) The percentage of symbionts covering host surface (host coverage) was measured with

Gimp 2.8 software after manual segmentation of the bacteria, whereby all partially and total cells in the 70 μm$^2$ frame were included in the analysis. All other parameters were analyzed with AnalySIS® program (Soft Imaging System GmbH, Münster, Germany), for each oral and aboral part of the microzooids up to 70 cells each, whereby cells were selected in a clockwise helical pattern: 3) length, 4) width, and 5) frequency of dividing cells (FDC). Dividing cells were defined as bacteria showing an invagination but not a clear intervening zone between cells [40]. 6) Cell volume was calculated from length and width data considering each cell as a cylinder plus two hemispheres [41]. 7) The cell elongation factor (EF), the ratio of length to width, was calculated for each cell [42]. The larger the EF the more rod-shaped the cells are, while cocci have an EF of approximately 1.

Statistical analyses were conducted with R [34] on data comparing three colonies kept at sulfide starvation for 48 h and three colonies collected *in situ* in 2012 (S2 Table). Because the Shapiro-Wilk tests performed for all parameters for each part of each microzooid showed deviations from normality, we used the Wilcoxon-Mann-Whitney test to evaluate differences within and between *in situ* and 48h oxic conditions.

## Results

### Host response to sulfide starvation compared to sulfidic conditions

*In situ* collections to investigate the host's reaction to experimental oxic conditions (so-called sulfide starvation) came from two nearby subtidal locations in the northern Adriatic Sea, the Strunjan mudflat and the Sv. Jernej estuary. No significant differences were found in colony size (Wilcoxon-Mann-Whitney test: p = 0.721, W = 941, S3A Fig) and in numbers of macrozooids between the locations (p = 0.487, W = 1931, S3B Fig). The number of swarmers released during sulfide starvation was highly variable, ranging from 0 to 21 swarmers per colony. However, the two populations did not differ significantly (p = 0.462, W = 1936, S3C Fig). The size of the swarmers also did not differ significantly between populations (p = 0.408, W = 1306, S3D Fig). We have, therefore, merged the data from both locations for further analyses and for comparisons with a population from Strunjan that we exposed to sulfidic conditions.

Monitoring the physicochemical parameters showed that the colonies were exposed to stable oxygen concentrations under sulfide starvation (97 ± 6% mean ± standard deviation in freshly supplied seawater, 99 ± 4% in the removed water 12 h later). In contrast, fluctuations alternated from almost anoxic (4 ± 4%), high sulfidic (448 ± 11 μmol L$^{-1}$ sulfide) conditions to oxic (89 ± 10%) and low sulfidic (6 ± 5 μmol L$^{-1}$ sulfide) conditions in 12 h intervals (S2 Table).

The mortality of the colonies under sulfide starvation showed a sigmoid pattern (bGLM: D$^2$ = 0.93), which increased sharply 24 h after the start of the experiment and increased less and less after 60 h (Fig 2A). All times are expressed as the upper bounds of the observation intervals. A similar mortality pattern (bGLM: D$^2$ = 0.97), but shifted to an increase about half a day later, was seen in colonies kept under sulfidic conditions. As a result, these colonies lived about half a day longer (LT$_{50}$ = 56 h, estimated standard error SE = 10 h) than those without sulfide (LT$_{50}$ = 44 h, SE = 11 h; Fig 2A).

In the sulfide starvation experiment significantly fewer swarmers were released per colony (median number per colony 1, IQR from 0 to 4, n = 120) compared to the sulfidic control (median number per colony 8, IQR from 6 to 11, n = 60; Wilcoxon-Mann-Whitney test comparing both experiments, p < 0.001, W = 6429). The mortality of swarmers from the sulfide starvation experiment also showed a sigmoid pattern similar to that of the colonies (bGLM: D$^2$ = 0.99). The calculated LT$_{50}$ was 39 h (SE = 9 h; Fig 2B). In contrast, the majority of swarmers (82%) who were kept in sulfidic seawater settled and grew into new colonies. Therefore,

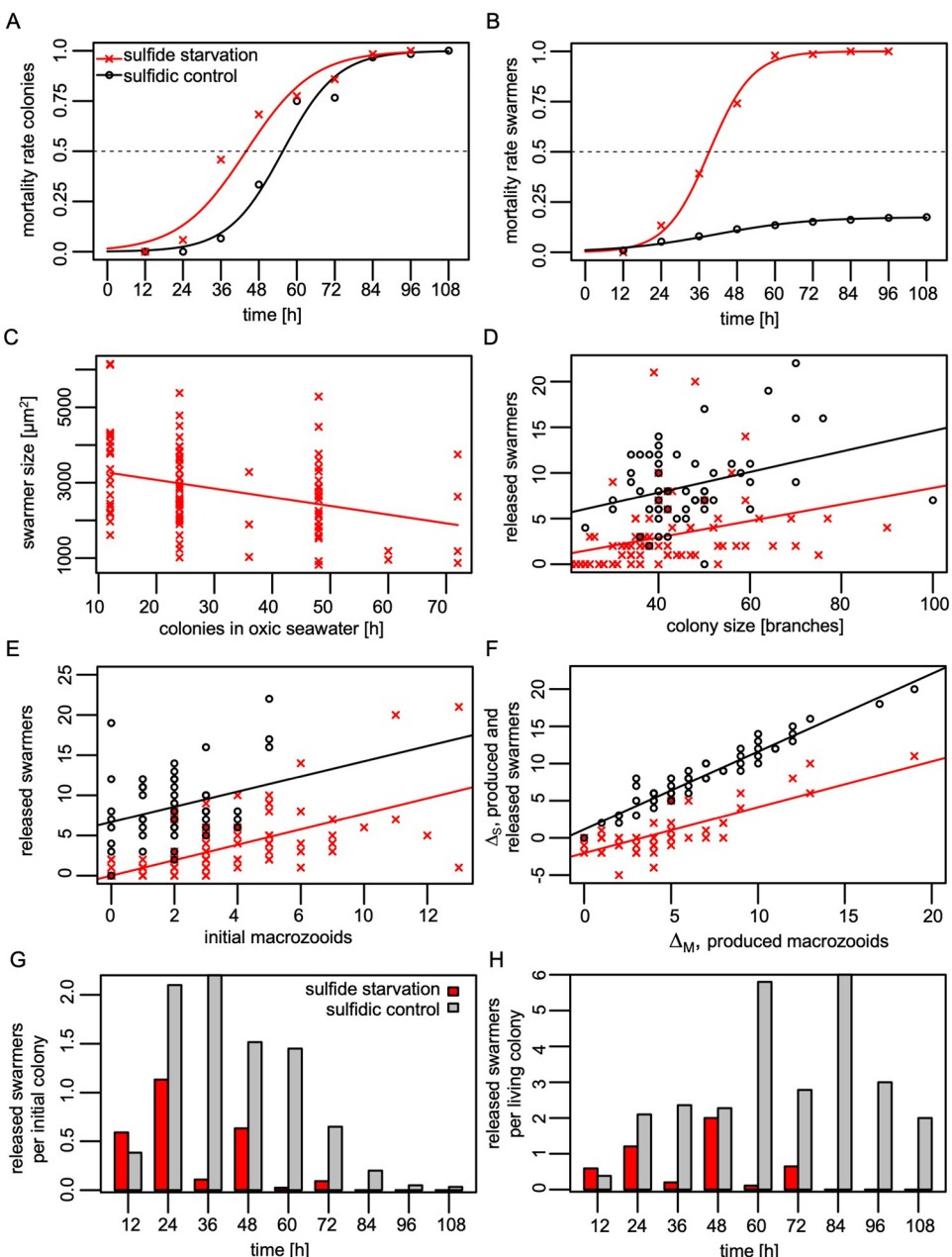

**Fig 2. Host response to sulfide starvation in comparison with a sulfidic control.** The sulfide starvation experiment is shown in red and the sulfidic control in black. (A) Binomial Generalized Linear Model of the mortality of the colonies given as the proportion of dead colonies in relation to the total number of colonies. $LT_{50}$ is indicated as the point of intersection with the dashed line. (B) Binomial Generalized Linear Model of the mortality of swarmers given as the proportion of dead swarmers in relation to the total number of swarmers. The x-axis marks the upper bound of the swarmer survival time after release from the colony. (C) Ordinary least squares regression model showing a negative correlation between swarmer size and time colonies spent under sulfide starvation before swarmer release. (D) Positive correlation between colony size and the number of released swarmers, both in the sulfide starvation experiment and in the sulfidic control. The slopes of both experiments were not significantly different. (E) The number of swarmers (released macrozooids) was positively correlated with the initial number of macrozooids, both in the sulfide starvation experiment and in the sulfidic control. The slopes of both experiments were not significantly different. $\Delta_S$ is defined as the difference between the number of released swarmers and the initial number of macrozooids. Positive values of $\Delta_S$ indicate the net number of additionally released swarmers, whereas negative values display the net number of macrozooids remaining on the colony. (F) $\Delta_S$, the net production and release of swarmers, is positively correlated with $\Delta_M$ the production of macrozooids in both sulfide starvation and sulfidic control experiments. The slopes of both experiments were significantly different. (G) At the population level, under sulfide

starvation less swarmers were released per initial colony than the sulfidic control in all time points except at 12h. They were also released in a shorter time period. (H) Each colony released less swarmers under sulfide starvation than in the sulfidic control also when only the colonies were considered which were still alive at each time-point.

the $LT_{50}$ of swarmers could not be calculated, as death of half of the swarmers was not reached (Fig 2B).

Swarmer size varied greatly between individuals in the sulfide starvation experiment (median = 26601 $\mu m^2$, IQR from 20737 to 35947 $\mu m^2$, n = 99). Measurements of size in 12-h-intervals according to the time a swarmer spent in the oxic water, showed no significant decrease (OLS: p = 0.077, F = 3, n = 99). However, swarmer size decreased in relation to the time colonies spent in oxic seawater (OLS: $r^2$ = 0.11, p < 0.001, F = 12, n = 99, Fig 2C). Colonies that spent more time without sulfide released significantly smaller swarmers. Whether this is the case under sulfidic conditions has not been investigated.

Although we tried to sample similarly sized colonies for both experiments, the ones used for the sulfide starvation experiment were slightly smaller (median 43 branches, IQR from 38 to 53, n = 85) than those used for the sulfidic experiment (median 44, IQR from 40 to 50, n = 60; Wilcoxon-Mann-Whitney test, p < 0.001, W = 1641). Nevertheless, the number of initial macrozooids per colony was not significantly different (sulfide starvation: median 2, IQR from 0 to 4, n = 120; sulfidic control: median 2, IQR from 1 to 3, n = 60; Wilcoxon-Mann-Whitney test, p = 0.444, W = 3849). The colony size correlated positively with the number of released swarmers in both experiments (sulfide starvation: OLS: $r^2$ = 0.12, p = 0.001, F = 11, n = 85; sulfidic control: OLS: $r^2$ = 0.13, p = 0.006, F = 8, n = 59; Fig 2D). Their slopes did not differ (analysis of covariance, p = 0.643, F = 0; Fig 2D).

In both treatments, a positive correlation was found between the number of initial macrozooids and the number of swarmers released (sulfide starvation: OLS $r^2$ = 0.47, p < 0.001, F = 106, n = 120; sulfidic control: OLS $r^2$ = 0.09, p = 0.017, F = 6, n = 60; Fig 2E). Their slopes did not differ significantly (analysis of covariance, p = 0.960, F = 0; Fig 2E). The median number of unreleased macrozooids at the end of the sulfide starvation experiment was 3 (IQR from 1 to 5, n = 80), significantly higher than the sulfidic control (median 1, IQR from 1 to 2, n = 59; Wilcoxon-Mann-Whitney test, p < 0.001, W = 3539).

To investigate whether the released swarmers came from macrozooids that were present at the beginning of the experiment or from macrozooids that developed during the experiment we calculated the production of new macrozooids ($\Delta_M$). In case of sulfide starvation, $\Delta_M$ showed a median of 3 (IQR from 1 to 5, n = 80). A significantly higher value was found in the sulfidic control experiment (median 6, IQR from 4 to 9, n = 60; Wilcoxon-Mann-Whitney test, p < 0.001, W = 1213). To investigate whether the colonies were able to release the additionally produced macrozooids ($\Delta_M$), we calculated $\Delta_S$, the number of macrozooids produced that were also released as swarmers during the experiment. In the sulfide starvation experiment $\Delta_S$ ranged from -5 to 11 with a median of 0 (IQR from -1 to 0, n = 71). This indicates high variability in colony performance: some colonies continued to produce and release these new swarmers ($\Delta_S$ > 0, n = 17), others produced no such swarmers ($\Delta_S$ = 0, n = 31), and others released only a few initial present macrozooids, while the remaining macrozooids died on the colony ($\Delta_S$ < 0, n = 23). In the sulfidic control experiment $\Delta_S$ ranged from 0 to 20 with a median of 8 (IQR from 5 to 10, n = 59; all $\Delta_S \geq 0$; Wilcoxon-Mann-Whitney test, p < 0.001, W = 317). In addition, both experiments showed a positive correlation between the macrozooids produced ($\Delta_M$) and the produced and released swarmers ($\Delta_S$) (Fig 2F). However, the slopes of both experiments were significantly different and indicate a different efficiency in production and release of swarmers from the colony (analysis of covariance, p < 0.001, F = 49;

Fig 2F). According to the sulfide starvation slope, the production of ten additional macro-zooids was needed in order to effectively release six more swarmers (OLS: $\Delta_S = -2.0 + 0.6 * \Delta_M$, $r^2 = 0.63$, $p < 0.01$, $F = 120$, $n = 71$; Fig 2F). In contrast, the slope of the sulfidic control showed that every newly produced macrozooid was also released as swarmer (OLS: $\Delta_S = 1.1 + 1.0 * \Delta_M$, $r^2 = 0.93$, $p < 0.001$, $F = 734$, $n = 59$). This indicates an overall higher efficiency in reproductive effort under sulfidic condition than without sulfide.

On population level, swarmer release was overall much lower under sulfide starvation and ended earlier than in the sulfidic control with the exception of the first time point at 12 h (Fig 2G). A similar picture emerged at the level of individual colonies (Fig 2H).

## Symbiont response to sulfide starvation

To study the change in symbiont coverage and colonization of other microbes in relation to host survival and time we repeated the sulfide starvation experiment with 10 to 20 colonies in different embryo dishes. We performed FISH with a symbiont-specific probe and a mixture of archaea and bacterial probes on semi-thin sections of some selected colonies, which were removed after each time point. All hosts survived for up to 24 h. At this point, the symbiont monolayer remained undisturbed in three of four colonies, similar in appearance to three colonies examined which were fixed immediately after collection from the field. The fourth colony suffered a small loss of symbionts.

After 48 h (n = 8; four live and four dead colonies) and after 72 h (n = 6; three live and three dead colonies) there was hardly any difference with regard to symbiont coverage on live and dead hosts. After 48 h, two out of the surviving four colonies showed little change in symbiont coverage (S4A–S4D Fig). One colony showed a large loss of symbionts, and one colony was aposymbiotic. Symbiont coverage on three out of four dead colonies was slightly disturbed, and the fourth dead colony suffered a great loss of symbionts. After 72 h two out of the three live colonies were aposymbiotic (S4E–S4H Fig), the third colony was severely disturbed and showed only very few symbionts. All three dead hosts were aposymbiotic.

The epigrowth of other microbes, including bacteria and archaea, began within the first 24 h. We found that microbial fouling originated mainly from the lower part of the colony (S4A–S4D Fig), sometimes overgrown by microbes in nature [11, 12]. Most of the time, other microbes colonized host surfaces after the symbiont was lost (Fig 3A–3D). In some cases, however, the symbiont monolayer was directly overgrown (Fig 3I–3L). Unspecific epigrowth was found in all colonies regardless of host viability.

In addition, unidentified microbes have been detected in a few microzooids. Some were limited to small spots and were most likely contained in food vacuoles (Fig 3A–3D). Others, however, filled the entire host cell, which we interpret as potential microbial infection (Fig 3E–3H). These individual host cell infections appeared to increase in number with time of incubation period and were randomly distributed within the colony. Since we did not find any branches, stalks, or clusters of infected microzooids, the infection did not appear to spread from one infected to neighboring microzooids.

In order to take into account the differences in symbiont morphology on the microzooids by means of SEM [12, 21], we differentiated between symbiont populations on the oral and aboral part of the microzooids freshly collected from the environment (Fig 4A) and compared them to those kept at oxic conditions for 48 h (Fig 4B–4G). Parts of the colony became covered with a mucus-like substance (Fig 4B) and/or other microbes (Fig 4F and 4G), which is in agreement with the FISH observations.

In freshly collected colonies (n = 3, Fig 4A), the microzooids were covered with a mono-layer of symbionts, with similar symbiont coverage values in the oral and in the aboral part

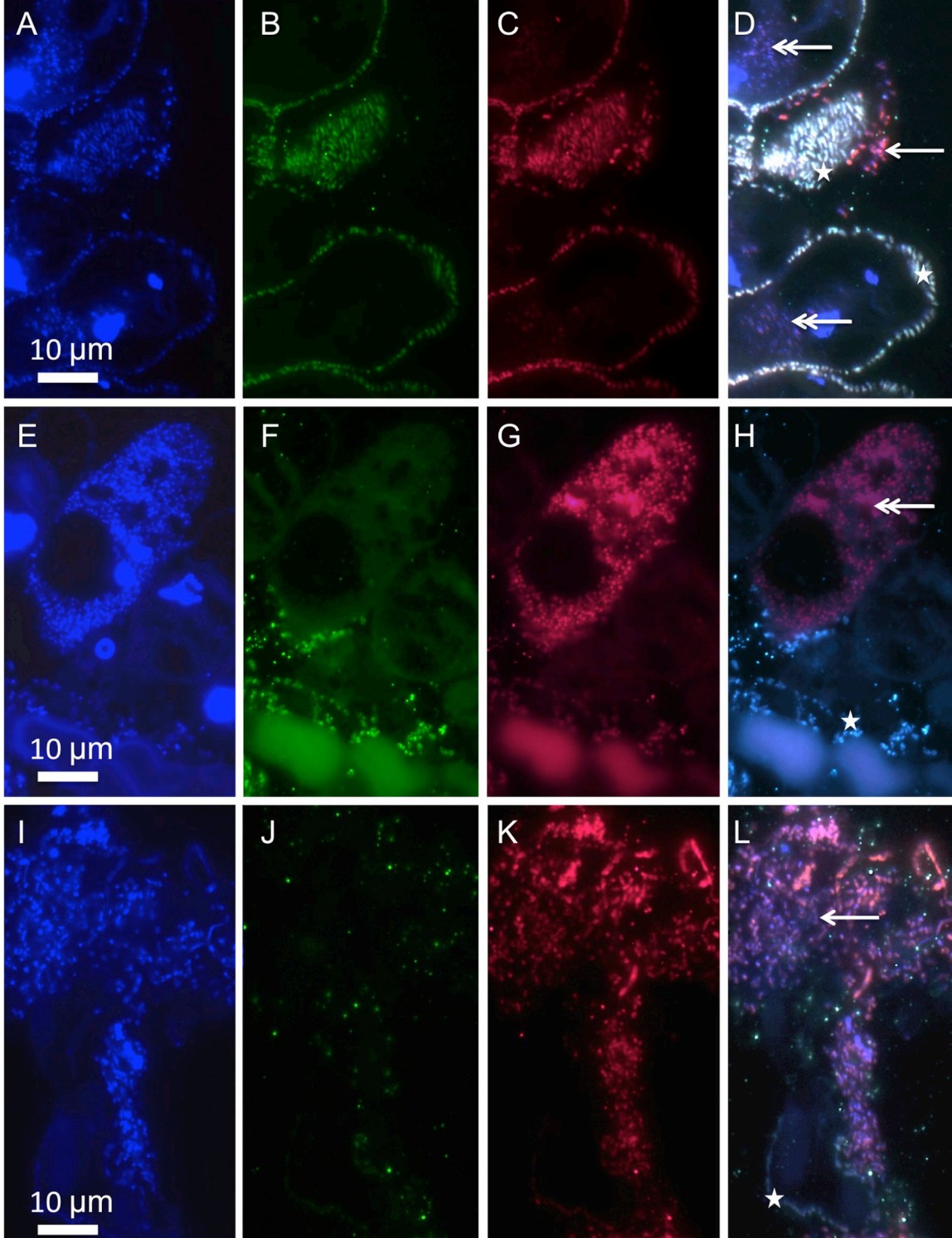

**Fig 3. Symbiont response to sulfide starvation applying FISH.** Symbionts (asterisk), other epibiotic microbes (arrow), and intracellular microbes in D confined to small areas most likely food vacuoles, in H filling the entire host cell most likely infection (double arrow); (A, E, I) DAPI staining (blue); (B, F, J) symbiont-specific probe (green); (C, G, K) EUB$_{mix}$ and Archaea probes (red); (D, H, L) composite of DAPI, symbiont-specific and EUB$_{mix}$/Archaea probes. A-D and I-L from live colony after 48 h, E-H from dead colony after 48 h.

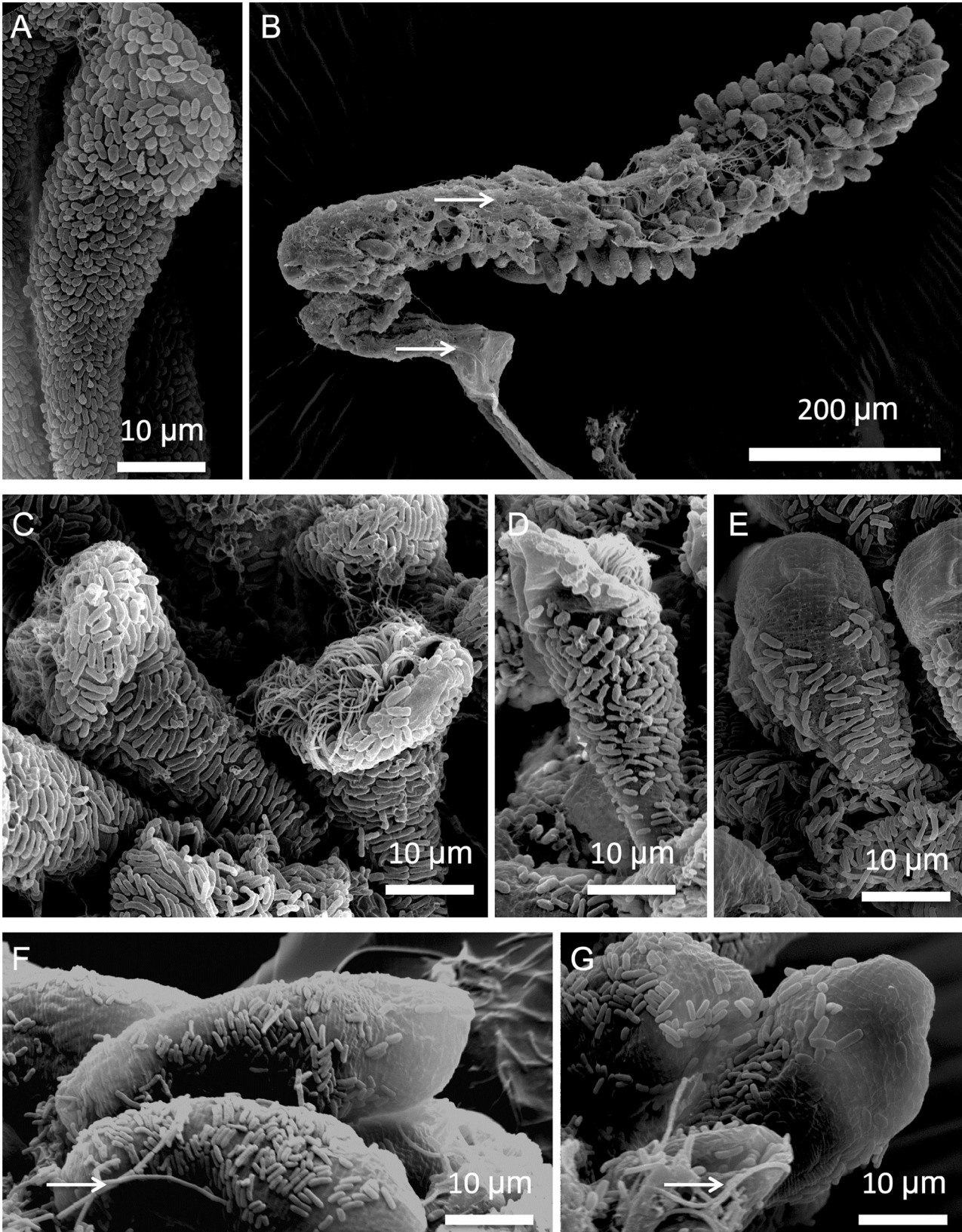

**Fig 4. Symbiont response to sulfide starvation applying SEM.** Microzooid from colony freshly collected from the environment (A), and several colonies after 48 h in oxic seawater (B-G); (B) overview of colony covered in part with mucus; (C-G) microzooids with symbionts fully covering the host, and with gradually less and less symbiont coverage; arrows point to very long rods most likely not symbionts.

**Table 1. SEM analyses of symbiont traits.**

| traits | *in situ* oral | *in situ* aboral | signinficance (W value) | 48 h oxic oral | 48 h oxic aboral | significance (W value) |
|---|---|---|---|---|---|---|
| coverage on host surface (%), n = 180 | 88.7 (85.9, 90.3) | 90.1 (88.3, 91.3) | n.s. (1302) | 10.4 (3.1, 36.2) | 62.7 (53.6, 79.2) | * (1799) |
| number of cells per 70 µm², n = 180 | 51.1 (43.5, 60.9) | 79.3 (68.5, 93.5) | * (1867) | 3.3 (1.1, 10.9) | 23.9 (16.3, 34,8) | * (1713) |
| frequency of dividing cells, FDC (%), n = 180 | 14.3 (12.9, 15.7) | 11.4 (10.0, 12.9) | * (323) | 7.7 (5.4, 9.1) | 9.1 (7.4, 11.4) | n.s. (1318) |
| length (µm), n = 9271 | 1.81 (1.53, 2.18) | 1.67 (1.39, 2.03) | * (3974591) | 2.43 (2.04, 2.90) | 2.56 (2.11, 3.06) | * (1256266) |
| width (µm), n = 9271 | 0.90 (0.78, 1.02) | 0.60 (0.52, 0.68) | * (878280) | 0.96 (0.76, 1.15) | 0.75 (0.64, 0.91) | * (675978) |
| individual cell volume (µm³), n = 9271 | 0.87 (0.61, 1.22) | 0.38 (0.27, 0.53) | * (1225584) | 1.41 (0.85, 2.10) | 0.96 (0.63, 1.45) | * (808464) |
| elongation factor, n = 9271 | 2.05 (1.65, 2.56) | 2.84 (2.29, 3.47) | * (7333468) | 2.56 (1.96, 3.35) | 3.35 (2.61, 4.16) | * (1590929) |

Values are shown as median and (Q25, Q75). Wilcoxon-Mann-Whitney test between oral and aboral parts: n.s. not significant, * 99% significance.

(Table 1). The host had significantly higher numbers of symbionts per unit of surface on the aboral part than on the oral part of microzooids (Table 1). Orally located symbionts were significantly longer and wider than on the aboral part; hence orally located symbionts had a larger volume than on the aboral part (Table 1). However, the cell elongation factor, calculated as length divided by width, showed that symbiont located aborally were more rod-shaped than those located orally (Table 1). The FDC of the symbiont population at the oral part was significantly higher than that at the aboral part (Table 1).

After 48 h in oxic seawater, the oral and the aboral symbiont populations differed significantly in all parameters at a 1% level of significance compared to freshly collected colonies (Table 1). Orally, symbiont coverage with a few symbionts was very low compared with a higher aboral coverage (Table 1). Orally localized symbionts were shorter and wider orally than aborally, had a higher cell volume and a lower elongation factor (Table 1).

The symbiont coverage on both parts of the microzooids changed dramatically within 48 h in oxic seawater. In comparison with freshly collected colonies, significantly lower symbiont coverage (Fig 5A and 5B) and symbiont number were observed on the oral and aboral parts of the microzooids (Fig 5C and 5D). Symbionts on both parts of the microzooids significantly increased in volume (Fig 5E and 5F) and became significantly more rod-shaped (Fig 5G and 5H). FDC was also significantly lower after 48 h (Table 1, Fig 5I and 5J) compared to freshly collected colonies.

## Discussion

While maintenance of host-microbe mutualism over a host generation requires finely tuned exchange of goods and services between partners, persistence over ecological time scales requires reproduction prior host death and transmission of symbionts from one to next host generation [14, 43, 44]. In unstable environments like those inhabited by the giant ciliate mutualism, one of the greatest, naturally occurring threats is the cessation of sulfide flow. We have shown in a series of experiments that the association breaks down quickly when exposed to such sulfide deficiency conditions. Reproduction of the host colonies by swarmers was sustained until the host died in less than two days, albeit to a lesser extent than under sulfidic conditions, which resulted in many more swarmers released. Most notably, the mixture of supplied sulfide and oxygen in the control experiment resulted in the settlement of 82% of

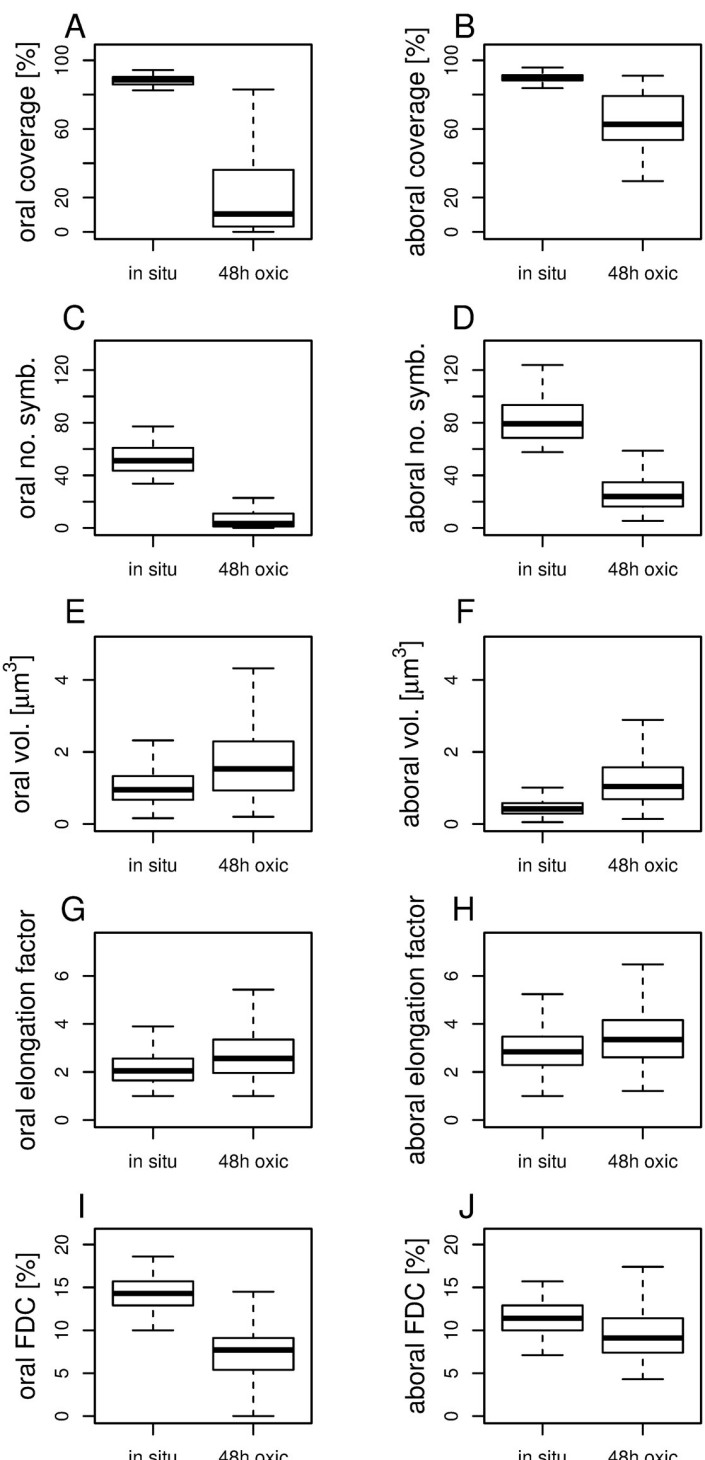

**Fig 5. Symbiont response to sulfide starvation compared to freshly collected colonies.** Box-and-whisker plots comparing orally (A, C, E, G, I) and aborally (B, D, F, H, J) located symbionts on microzooids of following parameters: percentage of symbiont coverage on host outer surface (A, B), number of symbionts per 70 μm² host surface area (C, D), volume of individual symbiont cells (E, F), cell elongation factor (ratio of symbiont length to width) (G, H), and frequency of dividing cells (FDC, ratio of dividing to total symbiont cells) (I, J). The box in the box-and-whisker plots shows the interquartile range with the median. The whiskers extend to the most extreme data points that are no more than 1.5 times the interquartile range from the box. All data were compared with the Wilcoxon-Mann-Whitney test and proved significantly different (99% significance) between *in situ* conditions and 48 h oxic experiments.

swarmers and growth into viable colonies. Symbionts lacking sulfide showed changes in the morphology on the host and reduced division within two days compared to *in situ* values. Consequently, the loss of symbiont coverage resulted in more or less aposymbiotic hosts, which were often overgrown by unspecific microbes.

Our experiments show that the host's efforts to develop propagules under sulfide starvation continued until host death. This was shown by the fact that swarmers came not only from macrozooids that were already present on the colony at the beginning of the experiment, but also from macrozooids that had developed during the experiment. Since not all of the macrozooids that were initially present and not all of the newly developed macrozooids were released, the total number of swarmers released corresponds roughly to the number of macrozooids initially present in the sulfide starvation experiment. Although the number of propagules was much fewer than those that developed under sulfidic conditions and never developed into new colonies, these results clearly indicate the importance of macrozooid production until the host dies. The fact that these swarmers never settled was not unexpected, as previous studies in the field [27] and in the lab [31] showed that sulfide is the settlement signal.

A total of 310 swarmers from 120 colonies (a median of one swarmer per colony) were released under sulfide starvation, with more than half of all swarmers leaving the colonies on the first day. The release did not cease until the host died. The swarmer size decreased significantly with time the colony was sulfide deficient. This indicates a trade-off between quality and quantity of offspring. In comparison, in the sulfidic control experiment 60 initial colonies released 515 swarmers (with a median of eight swarmers per colony), showing much greater success in releasing the offspring. This is also reflected in the significantly higher slope of the linear fit between produced macrozooids ($\Delta_M$) and produced and released swarmers ($\Delta_S$) in the sulfidic control than under sulfide starvation (Fig 2F). Whether the swarmers change in size with the amount of time the colonies spend in sulfidic water remains to be investigated.

Following the fate of the swarmers kept under sulfidic conditions, we found that out of 515 swarmers released from 60 colonies 425 first generation colonies developed releasing 7 second generation swarmers in 4.5 days. Even one of these second generation swarmers settled. A previous study under steady flow conditions with fully oxygenated seawater supplemented with low sulfide concentrations resulted in about 80 new colonies from 13 initial colonies in five days [21]. Although these cultures differed in concentrations of chemicals and flow versus stagnant seawater conditions, in both each colony produced between six to seven offspring. We note, however, that under flow many swarmers may have been flushed out before they were able to settle. Comparing our sulfide starvation experiment under stagnant oxic conditions with a previous oxic flow-through experiment [21] shows remarkable differences. While no swarmers settled in the former, in the latter 13 colonies produced 15 first generation colonies [21]. The cut-off seagrass leaf on which the initial colonies grew was placed in the flow-through chamber. We suspect that it might have leaked sulfide due to degradation and triggered swarmer settlement. Therefore, direct comparisons of culture conditions are difficult to interpret.

We showed that colony death was accelerated under oxic conditions compared to sulfidic conditions. We hypothesize that the lack of sulfide resulted in reduced diet for the host, which resulted in a shorter lifespan. Earlier studies showed that with prolonged sulfide starvation the carbon fixation in the symbiont ceases and then the release and uptake of organic carbon also stops [30]. Symbionts on colonies that were kept under sulfide starvation for 24 h before incubations with $^{14}$C or $^{13}$C labeled bicarbonate showed no carbon fixation and incorporation and no uptake into the host tissue took place [30]. The host diet under such oxic conditions is then reduced to direct ingestion of free-living microbes and symbionts [30]. We do not know yet, whether the changes in host nutrition alone or other as-yet-unknown benefits that the

sulfide-deficient symbiont did not provide resulted in a stressed host and accelerated death compared to colonies kept under similar but sulfidic conditions.

A remarkable phenotypic change occurred in the symbionts under sulfide starvation in just two days. Differences in morphology between symbionts on oral and aboral microzooid parts known in freshly collected colonies from the environment [21], were retained, but symbionts on both parts became more rod-shaped and grew larger compared to freshly collected colonies.

Surprisingly, FDC values show that the proliferation did not stop completely as expected, but was greatly reduced. In view of the fact that internal sulfur storage in the symbionts can only support carbon fixation for a very short time [23], the symbiont may switch to heterotrophic metabolism and therefore maintain proliferation. A recent study showed an upregulation of transporter genes, indicating heterotrophy under oxic conditions in *Cand*. Thiosymbion oneisti, the thiotrophic ectosymbiont of the marine nematode *Laxus oneistus* [45]. Although genes that support this function are present in the metagenome assembled genome of Thiobius (Espada-Hinojosa pers. obs.), it remains to be investigated whether they are expressed under such conditions.

FDC values did not differ significantly (with a significance level of 1%) in orally and aborally located symbionts who were exposed to oxic seawater (Table 1). This indicates similar abiotic conditions for the symbionts regardless of their position on the microzooids. These results are consistent with previous cultivation experiments under a steady flow of oxic seawater, but supplemented with sulfide, where oral and aboral symbionts also had similar FDC values [21]. In contrast, the FDC values of orally located symbionts from our *in situ* colonies freshly collected from wood were higher than those located aborally, confirming the results from colonies collected from degrading seagrass leaves and from vertical, overhanging rocks over seagrass debris [21].

Host–symbiont maintenance was clearly disturbed under sulfide starvation. The symbiont coverage on the host was significantly reduced compared to freshly collected colonies with a monolayer on the host. This may be due to reduced symbiont proliferation under sulfide deficiency in combination with a loss of symbionts due to ingestion by the host. Whether loss of symbionts can also be traced back to death and/or to escape into the environment remains to be investigated.

The disturbance of host–symbiont maintenance was also visible through microbial fouling on symbiont-free host surfaces or even on top of the symbiont. In freshly collected colonies epigrowth occurs from the lower part of the colonies [11, 12] in a manner similar to what we observed in stressed hosts. As these are the oldest parts of the colony, this may suggest that the age of the host plays a role in warding off microbial fouling under natural sulfidic as well as experimental sulfide starvation conditions. Alternatively or additionally, symbionts may contribute to the antimicrobial defense.

Not only the numbers of swarmers produced per colony, but also the survival of the swarmer is important as it sets the limits of dispersal in order to find a patchy, sulfide-leaking habitat for settlement. In addition, symbionts are transmitted vertically on the swarmer [11, 12]. However, under oxic conditions, the swarmer gradually becomes aposymbiotic. Almost 40% of swarmers lost their symbionts within 24 h and 100% of swarmers within 48 h [31]. With a swarmer $LT_{50}$ of 39 h and a considerable swimming speed of 5 mm s$^{-1}$ [46] a spread of approximately 700 m can be achieved if the swarmer swims in a straight line. This estimate does not take into account that the spread is also strongly influenced by currents. Both, the life span of the swarmer and the period of time to keep at least some of the symbionts are critical to maintaining mutualism by dispersal in search for the right sulfidic site to settle and establish a new colony.

In summary, our experiments show that the beneficial interactions between *Zoothamnium niveum* and its only symbiont *Cand.* Thiobius zoothamnicola is quickly disturbed under stressful oxic conditions without sulfide. As expected, colonies die quickly in less than two days. Importantly, we observed that they continue to produce propagules until death. Symbionts are also quickly affected, changing their morphology and slowing down division. Now that the principal mode of stress response is known, we can begin to decipher the underlying mechanisms of changes in physiology and interactions at the molecular level.

## Supporting information

**S1 Fig. Scheme of colony.** The colony is composed of a stalk with alternate branches and three different cell types–terminal zooids for division, microzooids for nutrition, and macrozooids for asexual reproduction. The size of the colony is counted in number of branches. A colony with initial macrozooids present at the start of the experiment and remaining macrozooids at the end of experiment is shown. During experimental time the release of swarmers was also counted.
(TIF)

**S2 Fig. Time schedule of the sulfide starvation experiment.** Colonies (n = 120) were monitored every 12 h (horizontal time line). Released swarmers from each of this time points were divided in 4 cohorts (A, B, C, D; vertical time line).
(TIF)

**S3 Fig. Comparison of colonies from Sv. Jernej and Strunjan.** All Wilcoxon-Mann-Whitney tests fail to reject the null hypothesis of equal medians: (A) colony size (p = 0.72), (B) number of initial macrozooids per colony (p = 0.49), (C) number of swarmers (released macrozooids) per colony (p = 0.46) and (D) size of swarmers (p = 0.41).
(TIF)

**S4 Fig. FISH micrographs colonies.** Colony alive after 48 h (A) DAPI staining (blue), (B) symbiont-specific probe (green), (C) $EUB_{mix}$ and Archea probes (red) (D) composite of A, B, C; note the increase in microbial fouling from top to bottom. Colony alive after 72 h with very few symbionts left (E) DAPI staining (blue), (F) symbiont-specific probe (green), (G) $EUB_{mix}$ and Archea probes (red), (H) composite of E, F and G.
(TIF)

**S1 Table. Collections and abiotic parameters measured prior collection at wood surface.** Samples listed according to type of experiment, applied technique, and time series of experiment, site, date of collection, number of wood, and abiotic parameters: depth, temperature, salinity, and pH. Abiotic parameters were measured using a Multi 340i sensor WTW.
(DOCX)

**S2 Table. Abiotic parameters measured at the start (and at the end) of experiments.** Abiotic parameters: temperature, salinity, pH, and oxygen and sulfide concentrations (mean ± standard deviation). Temperature, salinity, pH were measured using a Multi 340i sensor WTW. Oxygen concentration was measured using a PreSenS Flow-through Cell FTC-PSt3. Sulfide concentration was measured photometrically according to Cline (1969).
(DOCX)

## Acknowledgments

We would like to acknowledge the Marine Biology Station Piran (Slovenia) for their hospitality. EM work was performed at the Core Facility Cell Imaging and Ultrastructure Research,

University of Vienna. We would like to thank a reviewer for the highly valuable input and suggestions.

## Author Contributions

**Formal analysis:** Salvador Espada-Hinojosa, Judith Drexel, Julia Kesting, Edwin Kniha, Iason Pifeas, Lukas Schuster, Jean-Marie Volland, Helena C. Zambalos, Monika Bright.

**Funding acquisition:** Monika Bright.

**Investigation:** Salvador Espada-Hinojosa, Judith Drexel, Julia Kesting, Edwin Kniha, Iason Pifeas, Lukas Schuster, Jean-Marie Volland, Helena C. Zambalos, Monika Bright.

**Methodology:** Salvador Espada-Hinojosa, Judith Drexel, Julia Kesting, Edwin Kniha, Iason Pifeas, Lukas Schuster, Jean-Marie Volland, Helena C. Zambalos.

**Project administration:** Monika Bright.

**Supervision:** Monika Bright.

**Visualization:** Judith Drexel, Julia Kesting, Edwin Kniha, Iason Pifeas, Lukas Schuster, Helena C. Zambalos.

**Writing – original draft:** Salvador Espada-Hinojosa, Monika Bright.

**Writing – review & editing:** Salvador Espada-Hinojosa, Edwin Kniha, Lukas Schuster, Jean-Marie Volland, Monika Bright.

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
