## [Decision Letter · Decision Letter 0]

10 Aug 2021

PONE-D-21-21486

Host-symbiont stress response to lack-of-sulfide in the giant ciliate mutualism

PLOS ONE

Dear Dr. Espada-Hinojosa,

Thank you for submitting your manuscript to PLOS ONE. After careful consideration, we feel that it has merit but does not fully meet PLOS ONE’s publication criteria as it currently stands. Therefore, we invite you to submit a revised version of the manuscript that addresses the points raised during the review process.

We look forward to receiving your revised manuscript.

Kind regards,

Marcos Pileggi, Ph.D

Academic Editor

PLOS ONE

Journal Requirements:

“Funding came from two Austrian Science Fund projects FWF P24565 B22 and FWF 32197 granted to MB. (https://www.fwf.ac.at)”

Reviewers' comments:

Reviewer's Responses to Questions

**Comments to the Author**

1. Is the manuscript technically sound, and do the data support the conclusions?

Reviewer #1: Partly

Reviewer #2: Partly

2. Has the statistical analysis been performed appropriately and rigorously? 

Reviewer #1: Yes

Reviewer #2: Yes

3. Have the authors made all data underlying the findings in their manuscript fully available?

Reviewer #1: Yes

Reviewer #2: Yes

4. Is the manuscript presented in an intelligible fashion and written in standard English?

Reviewer #1: Yes

Reviewer #2: No

5. Review Comments to the Author

Reviewer #1: The paper is descriptive. It is primary research. There are lots of data and they are thoroughly analyzed. I believe that it meets the criteria for publication in PLOS One except for some statements not supported by data. I recommend that the text be revised to qualify more carefully statements and conclusions about why the colonies died.

My problems are with the statements (conclusions) that the ciliates and bacteria died because conditions were made oxic and sulfide was removed. Evidently, when taken into the laboratory and placed into well plates, the organisms invariably died. The cause is uncertain. There were no controls showing that the colonies could be kept alive if conditions were sulfidic and/or hypoxic. There are a number of alternative reasons that the colonies died. Failure to keep the colonies alive is puzzling because Rinke et al., 2007, described growing them in the lab, and Prof. Bright was a coauthor on that publication.

It is unclear to me how long the collections were held in the aquarium and how they fared. That may answer my concerns.

In my initial reading, I wondered: what exactly are microzooids, macrozoids, swarmers, colonies, and branches. It took me a while to realize that macrozooids, swarmers, and propagules are all nearly the same thing. “Symbiont” confused me; the problem is, in my mind, that both bacteria and ciliates are symbionts. I finally understood “symbionts” to be the bacteria. Some of these terms were defined in various places in the text and figure captions, but it would have been helpful to me to have terms defined at the beginning of the paper. A labeled figure would be best, showing a colony and how large it is, such as Fig. S2. I was not familiar this association and had to do several days of background reading. Bright(2019) was particularly helpful.

I suggest “presumed propagules” because they have not been shown to be viable. That is particularly true of the last ones to be released which are smaller and not known to carry bacteria.

In this symbiosis, the presumption is that the bacteria are transferring organic compounds to the ciliate. But the way in which bacteria completely cover the ciliate suggests that the bacteria are getting something from the ciliates. That would be consistent with the statement that the bacteria cannot be cultured separate from the ciliate. The younger ciliates are phagotrophic filter feeders, and might be transferring nutrients to the bacteria.

Throughout, the authors write “sulfur” where it should be “sulfide”. Sulfur is S(0) such as S(8), and sulfide is H2S and HS-.

L 28, please explain why this is an example of an r-strategy species. I ask that because some swarmers were retained on the colony until conditions became stressful. My understanding of an r-strategy would be to release the swarmers immediately for more offspring and a shortened reproductive cycle. Some trees retain their seeds until after there has been a fire, and I don’t believe that is an r-strategy. It is “serotiny”.

L 30, for “symbionts” I suggest using the word “bacteria”. The ciliates are symbionts too.

L 54, sentence needs work. Which organism is taking up nutrients? Since they live in a rich detrital environment, it is possible that both symbionts take up organics.

L 61, why “abiotic”? That is confusing because the text before was just describing biological sulfide production.

L 62, initially I read the list to be examples of “more suitable habitats.” This sentence could use work.

L 144, I had to read this sentence several times. Just, “filtered sea water” would be best, and describe earlier how it is filtered.

L 148, do not start a sentence with an Arabic number. If unavoidable, it should be spelled out. In any event, this sentence needs work because it can be read to mean 60 colonies in each well. And are colonies entire “stalks”? This could be made clear with a diagram and definitions of terms up front.

L 151, “prior” should be “at”. There is a missing word in this sentence.

L 155, pooled with water from other wells? It could be more clear.

L 190 “anymore until”, replace with “at”.

L 308, “maximal” is not right. Maybe, “at least”.

L 311, please mark the start of figure captions more clearly. I was confused on the initial reading.

L 457, “sign”-- I think means “significance”, but don’t leave the reader guessing. Punctuation could make it more clear.

Fig. 4, please describe briefly what the box and whisker plots show. For example, are the whiskers the total range or central 90%.

L 558, finding bacterial DNA in the water does not mean that bacteria were necessarily released. If they are disintegrating DNA will be released.

L 577, regarding maximum dispersal distances, that is affected mostly by currents.

To Salvador and the other authors I apologize for any incorrect readinds. It is well written and interesting material. I tend to get excited and too deeply involved.

Reviewer #2: Unfortunately the manuscript is not ready for review in its current form. The authors should review the manuscript for unusual syntax and unnecessarily wordy and convoluted sentence structure. These failings place an undue burden on the reviewer and make the job of reviewing the manuscript extremely difficult. Fortunately, the writing was mostly inteligible in the methods section and, based on this, it appears that the methods are sound. Unfortunately, the other sections are so poorly written that I gave up on my review after several hours of sentence-by-sentence translation into intelligible English. I am unable to determine whether the conclusions are supported by the data in this poorly written manuscript, although I suspect that they are. A complete revision is needed. Additionally a diagram and text description of the life cycle of the host and symbionts would make the manuscript more accessible to a general scientific audience. In summary, the study appears to relatively simple and straightforward in design and execution. It is quite possible that the research is of acceptable quality for publication. However, the manuscript is simply not ready for review.

6. PLOS authors have the option to publish the peer review history of their article (what does this mean?). If published, this will include your full peer review and any attached files.

Reviewer #1: No

Reviewer #2: No

---

## [Author Response · Author response to Decision Letter 0]

16 Nov 2021

Reviewer #1: The paper is descriptive. It is primary research. There are lots of data and they are thoroughly analyzed. I believe that it meets the criteria for publication in PLOS One except for some statements not supported by data. I recommend that the text be revised to qualify more carefully statements and conclusions about why the colonies died.

My problems are with the statements (conclusions) that the ciliates and bacteria died because conditions were made oxic and sulfide was removed. Evidently, when taken into the laboratory and placed into well plates, the organisms invariably died. The cause is uncertain. There were no controls showing that the colonies could be kept alive if conditions were sulfidic and/or hypoxic. There are a number of alternative reasons that the colonies died. Failure to keep the colonies alive is puzzling because Rinke et al., 2007, described growing them in the lab, and Prof. Bright was a coauthor on that publication.

We thank the reviewer for his comment pointing to the absence of a control with sulfide/low oxygen. Based on this suggestion we went back to the field and performed this control. 

Rinke et al. succeeded in cultivating the ciliate with its symbiont only in sulfide and oxygen. The attempt to culture in oxic seawater led to the initial colonies on a piece of seagrass leaf to release swarmers that settled and grew but the population died. The puzzling thing was that they even settled because we tested in another paper (Bright et al. 2019) the cue for settlement and found the out of 3000 swarmers only 1% settled without sulfide. So we believe that the seagrass leaf might have started to degrade and released sulfide. Moreover, when designing the experiment for this study we knew already that they will die fast but what we tried to investigate was how they respond in terms of macrozooid development and release of swarmers and what happens with their symbionts. 

In contrast to Rinke et al where cultivations were performed in mini flow through aquaria with a set of colonies together, we decided to perform this study under non-flow through conditions: 1) not to lose swarmers through the outlet of the chamber under flow, 2) to be able to follow life/dead condition of each colony and each swarmer, 3) to follow which colony releases how many swarmers, 4) to mimic better the switch from fluctuating high sulfide/anoxic condition to pure oxic conditions without sulfide as it found in nature. 

Therefore, we included in the text now: 

(old line 113, new line 133, track changes line 227): "... (in short sulfide starvation)"

(old line 116, new line 136, track changes line 230): "For comparison we also performed a sulfidic control experiment."

(old line 140, new line 163, track changes line 291): "... for the sulfide starvation experiment and 23 days for the sulfidic control experiment."

(old line 147, new line 170, track changes line 299): "... to sulfide starvation compared to sulfidic condition"

(old line 148, new line 172, track changes line 301): "... for the sulfide starvation experiment in 2015. For comparison, 60 colonies were sampled for the sulfidic control experiment at Strunjan in 2021."

(old line 149, new line 175, track changes line 305): "... (sulfide starvation experiment). For the sulfidic control we added sodium sulfide to the filtered seawater at an average final concentration of 448 µmol L-1. "

(old line 155, new line 183, track changes line 312): "... replaced by new filtered oxic seawater (sulfide starvation experiment) or filtered sulfidic seawater (sulfide control experiment). The removed water was ..."

(old line 157, new line 188, track changes line 325): "Additionally, sulfide concentration was measured photometrically according to Cline [32] in a few randomly chosen wells in the sulfidic control experiment in the newly prepared and in the removed water."

(old line 159, new line 193, track changes line 330): "... sulfide starvation experiment: n = 85; sulfidic control experiment: n = 60 ..."

(old line 173, new line 209, track changes line 346): "In both, the sulfide starvation and sulfidic control experiments, the mortalities ..."

(old line 193, new line 230, track changes line 374): "... sulfide starvation"

(old line 291, new line 306, track changes line 502): "... sulfide starvation compared to sulfidic condition"

(old line 292, new line 308, track changes line 505): "... (so-called sulfide starvation) ..."

(old line 303, new line 322, track changes line 519): "In contrast, fluctuations alternated from almost anoxic (4 ± 4 %), high sulfidic (448 ± 11 µmol L-1 sulfide) conditions to oxic (89 ± 10 %) and low sulfidic (6 ± 5 µmol L-1 sulfide) conditions in 12 h intervals. "

(old line 306, new line 328, track changes line 553): "A similar mortality pattern (bGLM: D2 = 0.97), but shifted to an increase about half a day later, was seen in colonies kept under sulfidic conditions. As a result, these colonies lived about half a day longer (LT50 = 56 h, estimated standard error SE = 10 h) than those without sulfide (LT50 = 44 h, SE = 11 h; Fig 2 A)."

(old line 311, new line 334, track changes line 560): "... sulfide starvation in comparison with a sulfidic control. The sulfide starvation experiment is shown in red and the sulfidic control in black."

(old line 320, new line 344, track changes line 570): "..., both in the sulfide starvation experiment and in the sulfidic control. The slopes of both experiments were not significantly different."

(old line 322, new line 347, track changes line 573): "... , both in the sulfide starvation experiment and in the sulfidic control. The slopes of both experiments were not significantly different "

(old line 329, new line 353, track changes line 593): "... in both sulfide starvation and sulfidic control experiments. The slopes of both experiments were significantly different."

(old line 329, new line 355, track changes line 594): "At the population level, under sulfide starvation less swarmers were released per initial colony than the sulfidic control in all time points except at 12h. They were also released in a shorter time period."

(old line 330, new line 357, track changes line 597): "Each colony released less swarmers under sulfide starvation than in the sulfidic control also when only the colonies were considered which were still alive at each time-point."

(old line 332, new line 362, track changes line 602): "In the sulfide starvation experiment significantly fewer swarmers were released per colony (median number per colony 1, IQR from 0 to 4, n = 120) compared to the sulfidic control (median number per colony 8, IQR from 6 to 11, n = 60; Wilcoxon-Mann-Whitney test comparing both experiments, p < 0.001, W = 6429)."

(old line 333, new line 368, track changes line 608): "In contrast, the majority of swarmers (82%) who were kept in sulfidic seawater settled and grew into new colonies. Therefore, the LT50 of swarmers could not be calculated, as death of half of the swarmers was not reached (Fig 2 B)."

(old line 341, new line 377, track changes line 642): "Whether this is the case under sulfidic conditions has not been investigated."

(old line 342, new line 379, track changes line 644): "Although we tried to sample similarly sized colonies for both experiments, the ones used for the sulfide starvation experiment were slightly smaller (median 43 branches, IQR from 38 to 53, n = 85) than those used for the sulfidic experiment (median 44, IQR from 40 to 50, n = 60; Wilcoxon-Mann-Whitney test, p < 0.001, W = 1641). Nevertheless, the number of initial macrozooids per colony was not significantly different (sulfide starvation: median 2, IQR from 0 to 4, n = 120; sulfidic control: median 2, IQR from 1 to 3, n = 60; Wilcoxon-Mann-Whitney test, p = 0.444, W = 3849). The colony size correlated positively with the number of released swarmers in both experiments (sulfide starvation: OLS: r2 = 0.12, p = 0.001, F = 11, n = 85; sulfidic control: OLS: r2 = 0.13, p = 0.006, F = 8, n = 59; Fig 2 D). Their slopes did not differ (analysis of covariance, p = 0.643, F = 0; Fig 2 D)."

(old line 342, new line 390, track changes line 655): "In both treatments, a positive correlation was found between the number of initial macrozooids and the number of swarmers released (sulfide starvation: OLS r2 = 0.47, p < 0.001, F = 106, n = 120; sulfidic control: OLS r2 = 0.09, p = 0.017, F = 6, n = 60; Fig 2 E). Their slopes did not differ significantly either (analysis of covariance, p = 0.960, F = 0; Fig 2 E). The median number of unreleased macrozooids at the end of the sulfide starvation experiment was 3 (IQR from 1 to 5, n = 80), significantly higher than the sulfidic control (median 1, IQR from 1 to 2, n = 59; Wilcoxon-Mann-Whitney test, p < 0.001, W = 3539)."

(old line 366, new line 400, track changes line 695): "In case of sulfide starvation, ∆M showed a median of 3 (IQR from 1 to 5, n = 80). A significantly higher value was found in the sulfidic control experiment (median 6, IQR from 4 to 9, n = 60; Wilcoxon-Mann-Whitney test, p < 0.001, W = 1213)."

(old line 371, new line 407, track changes line 702): "This indicates high variability in colony performance: some colonies continued to produce and release these new swarmers (∆S > 0, n = 17), others produced no such swarmers (∆S = 0, n = 31), and others released only a few initial present macrozooids, while the remaining macrozooids on the colony died on the colony (∆S < 0, n = 23). In the sulfidic control experiment ∆S ranged from 0 to 20 with a median of 8 (IQR from 5 to 10, n = 59; all ∆S ≥ 0; Wilcoxon-Mann-Whitney test, p < 0.001, W = 317). In addition, both experiments showed ..."

(old line 376, new line 415, track changes line 710): "However, the slopes of both experiments were significantly different and indicate a different efficiency in production and release of swarmers from the colony (analysis of covariance, p < 0.001, F = 49; Fig 2 F)."

(old line 380, new line 420, track changes line 715): "In contrast, the slope of the sulfidic control showed that every newly produced macrozooid was also released as swarmer (OLS: ∆S = 1.1 + 1.0 * ∆M, r2 = 0.93, p < 0.001, F = 734, n = 59). This indicates an overall higher efficiency in reproductive effort under sulfidic condition than without sulfide.

On population level, swarmer release was overall much lower under sulfide starvation and ended earlier than in the sulfidic control with the exception of the first time point at 12 h (Fig 2 G). A similar picture emerged at the level of individual colonies (Fig 2 H)."

(old line 499, new line 544, track changes line 1094): "Reproduction of the host colonies by swarmers was sustained until the host died in less than two days, albeit to a lesser extent than under sulfidic conditions, which resulted in many more swarmers released. Most notably, the mixture of supplied sulfide and oxygen in the control experiment resulted in the settlement of 82% of swarmers and growth into viable colonies."

(old line 509, new line 560, track changes line 1109): "Although the number of propagules was much fewer than those that developed under sulfidic conditions and never developed into new colonies, these results ..."

(old line 517, new line 569, track changes line 1159): "In comparison, in the sulfidic control experiment 60 initial colonies released 515 swarmers (with a median of eight swarmers per colony), showing much greater success in releasing the offspring. This is also reflected in the significantly higher slope of the linear fit between produced macrozooids (∆M) and produced and released swarmers (∆S) in the sulfidic control than under sulfide starvation (Fig 2 F). Whether the swarmers change in size with the amount of time they spend in sulfidic water remains to be investigated.

Following the fate of the swarmers kept under sulfidic conditions, we found that out of 515 swarmers released from 60 colonies 425 first generation colonies developed releasing 7 second generation swarmers in 4.5 days. Even one of these second generation swarmers settled. A previous study under steady flow conditions with fully oxygenated seawater supplemented with low sulfide concentrations resulted in about 80 new colonies from 13 initial colonies in five days [21]. Although these cultures differed in concentrations of chemicals and flow versus stagnant seawater conditions, in both each colony produced between six to seven offspring. We note, however, that under flow many swarmers may have been flushed out before they were able to settle. Comparing our sulfide starvation experiment under stagnant oxic conditions with a previous oxic flow-through experiment [21] shows remarkable differences. While no swarmers settled in the former, in the latter 13 colonies produced 15 first generation colonies [21]. The cut-off seagrass leaf on which the initial colonies grew was placed in the flow-through chamber. We suspect that it might have leaked sulfide due to degradation and triggered swarmer settlement. Therefore, direct comparisons of culture conditions are difficult to interpret.

We showed that colony death was accelerated under oxic conditions compared to sulfidic conditions."

(old line 527, new line 602, track changes line 1211): "... and accelerated death compared to colonies kept under similar but sulfidic conditions."

It is unclear to me how long the collections were held in the aquarium and how they fared. That may answer my concerns.

The colonies for the sulfide starvation experiment were kept in the large aquaria from some hours to five days. The sampling of the sulfidic control was done on July 21 and the experiment started on Aug 12. These colonies were grown in the large flow through aquaria for 22 days. 

We note that by switching from oxic flow-through conditions to adding high amounts of sulfide once a day and switching off the flow for some hours, we can keep the ‘culture’ alive as long as we decide. Thereby, the swarmers usually settle on the plastic surfaces of the aquaria in hundreds of colonies. This procedure we have done now every year since 2014. However, not being able to follow growth and death of each colony individually this way we performed experiments such as the one presented here. 

We changed the text: 

(old line 141, new line 162, track changes line 283): "... from immediately after collection up to five days later for the sulfide starvation experiment and 23 days for the sulfidic control experiment."

In my initial reading, I wondered: what exactly are microzooids, macrozoids, swarmers, colonies, and branches. It took me a while to realize that macrozooids, swarmers, and propagules are all nearly the same thing. 

We added a paragraph in the Introduction and new Figure 1 now to show the life cycle and all the terminology used.

Added paragraph:

(old line 91, new line 83, track changes line 155): "The vertical transmission of the ectosymbiont is through host propagules, called macrozooids, which are released as swarmers into the pelagial for dispersal. Once settled, the swarmer transforms into the terminal zooid and begins to produce the stalk and to divide, producing the terminal zooid of each branch. Nourishing microzooids are produced through division of the terminal zooid on each branch, increasing the length of the branch. Macrozooids develop at the base of the branch. These macrozooids leave the colony as soon as a ciliary band has formed."

(old line 99, new line 91, track changes line 174): "Fig 1. Life cycle of Zoothamnium niveum. The swarmers are the dispersal stage (1), and look for a sulfide source to settle. Once settled, the swarmer transforms into the terminal zooid at the top of the new colony and grows a stalk. Note that the white part of the stalk is overgrown by the symbiont, but the lower black part is aposymbiotic (2). The terminal zooid divides and produces the terminal zooids for each branch (3). The branch grows by divisions of the terminal zooid on the tip of the branch, creating microzooids and macrozooids, that eventually detach as swarmers (4). Light micrographs not to scale."

"Symbiont" confused me; the problem is, in my mind, that both bacteria and ciliates are symbionts. I finally understood "symbionts" to be the bacteria. Some of these terms were defined in various places in the text and figure captions, but it would have been helpful to me to have terms defined at the beginning of the paper. A labeled figure would be best, showing a colony and how large it is, such as Fig. S2. I was not familiar this association and had to do several days of background reading. Bright (2019) was particularly helpful. I suggest "presumed propagules" because they have not been shown to be viable. That is particularly true of the last ones to be released which are smaller and not known to carry bacteria.

We designed a new figure 1. 

We use the term propagule now to explain the asexually produced macrozooids turning into swarmers upon release. We also tried to explain now better that the propagules were not able to settle because the experiment was done in oxic seawater and the swarmers need sulfide to settle, which happened in 82% of swarmers that grew as colonies in the suggested new experiment of the reviewer with sulfide. 

We added:

(old line 91, new line 83, track changes line 155): "The vertical transmission of the ectosymbiont is through host propagules, called macrozooids, ..."

(old line 110, new line 130, track changes line 224): "Here, we followed the fate of large colonies and their propagules experimentally mimicking the waning of sulfide."

(old line 504, new line 553, track changes line 1102): "Our experiments show that the host’s efforts to develop propagules under sulfide starvation continued..."

(old line 509, new line 560, track changes line 1109): "Although the number of propagules was much fewer than those that developed under sulfidic conditions and never developed into new colonies, these results ..."

(old line 512, new line 562, track changes line 1152): "The fact that these swarmers never settled was not unexpected, as previous studies in the field [27] and in the lab [31] showed that sulfide is the settlement signal."

(old line 584, new line 657, track changes line 1421): "Importantly, we observed that they continue to produce propagules until death."

To clarify, we note that the symbiosis community uses the term symbiont(s) for the relatively smaller partner(s) and host for the larger partner regardless of absolute size (e.g. Douglas 2010). Therefore, in our case the bacterium is the symbiont and the ciliate is the host. This terminology is also used for other ciliates and their bacterial and archaeal symbionts such as Metopus and sulfate reducing bacteria and methanogen archaea. We are aware that ciliates can be symbionts of larger hosts, such as other peritrich ciliates on crustaceans but this is irrelevant for our study object. To clarify this point we changed the text to:

(old line 43, new line 40, track changes line 40): "... protist or animal hosts and thioautotrophic bacterial symbionts ..."

In this symbiosis, the presumption is that the bacteria are transferring organic compounds to the ciliate. But the way in which bacteria completely cover the ciliate suggests that the bacteria are getting something from the ciliates. That would be consistent with the statement that the bacteria cannot be cultured separate from the ciliate. The younger ciliates are phagotrophic filter feeders, and might be transferring nutrients to the bacteria.

The reviewer is assuming correctly. However, we have good evidence for the transfer of organic carbon from symbiont to host (Volland et al. 2018). What goods the host provides to the symbiont is not clear. We currently work on the publication of the symbiont genome and saw that the symbiont has several genes encoding for transporter uptake of organic molecules. This we plan to test in future. 

Throughout, the authors write "sulfur" where it should be "sulfide". Sulfur is S(0) such as S(8), and sulfide is H2S and HS-.

The reviewer is correct. We wrote twice sulfur-oxidizing bacteria because many microbial physiologists comprise in this phylogenetically diverse group all bacteria (and archaea) being able to oxidize any reduced sulfur species and are either autotrophic or heterotrophic. To clarify this point we use the term thioautotrophic now and explain this term:

(old line 21, new line 21, track changes line 21): "The mutualism between the thioautotrophic bacterial ectosymbiont …"

(old line 44, new line 41, track changes line 41): "… thioautotrophic bacterial symbionts depend on the presence of sulfide. These symbionts share the need for reduced sulfur species (e.g. exclusively sulfide or additionally thiosulfate) and …" 

Change from thiotrophic to thioautotrophic in:

old line 78, new line 70, track changes line 143; and

old line 80, new line 73, track changes line 145

L 28, please explain why this is an example of an r-strategy species. I ask that because some swarmers were retained on the colony until conditions became stressful. My understanding of an r-strategy would be to release the swarmers immediately for more offspring and a shortened reproductive cycle. Some trees retain their seeds until after there has been a fire, and I don't believe that is an r-strategy. It is "serotiny".

We agree with the reviewer and omit the sentence about r-strategy. We think, however, that also serotiny does not well describe what happens in this ciliate. We prefer not to use this term because from the new data obtained from the sulfidic experiment we clearly can see that during the last days of their life the oxic colonies released a median of one swarmer while the sulfidic ones released 8. Both started with 2 macrozooids. The oxic colonies end with 3 while the sulfidic one ended with one. This to us rather suggests that the sulfidic kept colonies fared better in terms of producing macrozooids and sending off swarmers. 

L 30, for "symbionts" I suggest using the word "bacteria". The ciliates are symbionts too.

We tried to make clear now that the symbionts are bacteria but prefer to continue to use symbiont for comparative reasons.

L 54, sentence needs work. Which organism is taking up nutrients? Since they live in a rich detrital environment, it is possible that both symbionts take up organics.

Yes this might be the case but we have no evidence for uptake in the bacterial symbionts. 

We clarified in the sentence now that the host takes up fixed organic carbon from the symbiont released immediately after fixation (Volland et al. 2018): 

Changed to: 

(old line 52, new line 48, track changes line 96): "In return, the symbionts nourish their hosts (see [3])."

L 61, why "abiotic"? That is confusing because the text before was just describing biological sulfide production.

We agree and omitted 'biological' in this sentence.

Changed to: 

(old line 61, new line 53, track changes line 102): "Upon changes in chemical conditions, ..."

L 62, initially I read the list to be examples of "more suitable habitats." This sentence could use work.

We changed the sentence to:

(old line 61, new line 53, track changes line 102): "Upon changes in chemical conditions, mobile animal hosts, e.g. stilbonematine nematodes, gutless oligochaetes, snails and bathymodiolin mussels, can migrate to more suitable habitats (see [3])." 

L 144, I had to read this sentence several times. Just, "filtered sea water" would be best, and describe earlier how it is filtered.

We follow the reviewer, remove the "0.2 µM" through the text and add the following sentence:

(old line 145, new line 166, track changes line 294): "Each colony was cut off the wood with a MicroPointTM Scissor and cleaned from debris by rinsing it in filtered seawater prior the experimental procedure. All seawater used for this and further procedures was filtered through a 0.2 μm Acrodisc® syringe filter."

L 148, do not start a sentence with an Arabic number. If unavoidable, it should be spelled out. In any event, this sentence needs work because it can be read to mean 60 colonies in each well. 

Corrected

(old line 148, new line 172, track changes line 301): "We used 60 colonies ..." 

And are colonies entire "stalks"? This could be made clear with a diagram and definitions of terms up front

We clarified the terminology now in the introduction and with a new figure 1. 

L 151, "prior" should be "at". There is a missing word in this sentence.

Corrected to:

(old line 150, new line 178, track changes line 307): "The number of macrozooids present on each colony was counted at the start of the experiment."

L 155, pooled with water from other wells? It could be more clear.

We changed to:

(old line 154, new line 183, track changes line 312): "Every 12 h about two-thirds of the water from each well was replaced by new filtered seawater. The removed water was pooled for measurements of temperature, salinity, pH, and oxygen concentration (S2 Table)." 

L 190 "anymore until", replace with "at".

We changed to:

(old line 187, new line 224, track changes line 367): "A positive ∆S value indicates additionally produced swarmers under oxic conditions during the experimental time frame, whereas a negative ∆S value indicates remaining macrozooids on the respective colony at the end of the experiment." 

L 308, "maximal" is not right. Maybe, "at least".

(old line 308, new line 331, track changes line 557): This sentence has been removed from the manuscript.

L 311, please mark the start of figure captions more clearly. I was confused on the initial reading.

Done

L 457, "sign"-- I think means "significance", but don't leave the reader guessing. Punctuation could make it more clear.

Corrected in the table to "significance". The caption of the table has been changed to: 

(old line 457, new line 504, track changes line 1014): "Values are shown as median and (Q25, Q75). Wilcoxon-Mann-Whitney test between oral and aboral parts: n.s. not significant, * 99% significance." 

Fig. 4, please describe briefly what the box and whisker plots show. For example, are the whiskers the total range or central 90%.

We added a sentence:

(old line 480, new line 529, track changes line 1048): "The box in the box-and-whisker plot shows the interquartile range with the median. The whiskers extend to the most extreme data points which are no more than 1.5 times the interquartile range from the box." 

L 558, finding bacterial DNA in the water does not mean that bacteria were necessarily released. If they are disintegrating DNA will be released.

We agree with the reviewer and remove this part of the manuscript. 

L 577, regarding maximum dispersal distances, that is affected mostly by currents.

We added a sentence:

(old line 575, new line 649, track changes line 1352): "This estimate does not take into account that the spread is also strongly influenced by currents. All these aspects, the life span of the swarmer and the period of time to keep at least some of the symbionts are critical to maintaining mutualism by dispersal in search for the right sulfidic site to settle and establish a new colony."

To Salvador and the other authors I apologize for any incorrect readings. It is well written and interesting material. I tend to get excited and too deeply involved.

We would to thank the reviewer very deeply for his efforts to help us to improve this manuscript. Special thanks to the suggestion to make the appropriate control. It was fortunate that when receiving the comments of the reviewers we were working in the field and were able to add some working days and do the control, although without measuring the size of the swarmers because we had no light microscope with camera. The added data did not change our interpretation but the text and the figure to highlight the differences for this part of the manuscript.

Reviewer #2: Unfortunately the manuscript is not ready for review in its current form. The authors should review the manuscript for unusual syntax and unnecessarily wordy and convoluted sentence structure. These failings place an undue burden on the reviewer and make the job of reviewing the manuscript extremely difficult. Fortunately, the writing was mostly inteligible in the methods section and, based on this, it appears that the methods are sound. Unfortunately, the other sections are so poorly written that I gave up on my review after several hours of sentence-by-sentence translation into intelligible English. I am unable to determine whether the conclusions are supported by the data in this poorly written manuscript, although I suspect that they are. A complete revision is needed. Additionally a diagram and text description of the life cycle of the host and symbionts would make the manuscript more accessible to a general scientific audience. In summary, the study appears to relatively simple and straightforward in design and execution. It is quite possible that the research is of acceptable quality for publication. However, the manuscript is simply not ready for review.

We apologize for the quality of the writing. We considerably corrected the text and simplified convoluted sentences and deleted unnecessary wording as the reviewer suggested. 

We added some more introduction paragraphs and a new figure showing all the terminology used for peritrich ciliates and their life cycle.

---

## [Decision Letter · Decision Letter 1]

6 Dec 2021

PONE-D-21-21486R1Host-symbiont stress response to lack-of-sulfide in the giant ciliate mutualismPLOS ONE

Dear Dr. Espada-Hinojosa,

Thank you for submitting your manuscript to PLOS ONE. After careful consideration, we feel that it has merit but does not fully meet PLOS ONE’s publication criteria as it currently stands. Therefore, we invite you to submit a revised version of the manuscript that addresses the points raised during the review process.

Although the reviewer's suggestions were accepted by the authors, some more corrections are still needed.

We look forward to receiving your revised manuscript.

Kind regards,

Marcos Pileggi, Ph.D

Academic Editor

PLOS ONE

Journal Requirements:

Reviewers' comments:

Reviewer's Responses to Questions

**Comments to the Author**

1. If the authors have adequately addressed your comments raised in a previous round of review and you feel that this manuscript is now acceptable for publication, you may indicate that here to bypass the “Comments to the Author” section, enter your conflict of interest statement in the “Confidential to Editor” section, and submit your "Accept" recommendation.

Reviewer #1: All comments have been addressed

2. Is the manuscript technically sound, and do the data support the conclusions?

Reviewer #1: (No Response)

3. Has the statistical analysis been performed appropriately and rigorously? 

Reviewer #1: Yes

4. Have the authors made all data underlying the findings in their manuscript fully available?

Reviewer #1: Yes

5. Is the manuscript presented in an intelligible fashion and written in standard English?

Reviewer #1: Yes

6. Review Comments to the Author

Reviewer #1: It's a good revision. I recommend acceptance. However, I ask the authors to consider the following suggestions.

line 81. Addition of Fig. 1 has helped me to understand what the symbiosis looks like, but for readers who are unfamiliar with it (such as me) zooids are still not well described upfront. My dictionary defines zooids as individual units of colony, but in the present case the description could be more precise. That would make the paper more accessible. Please consider something like: “Each colony consists of a central stalk of the giant ciliate Zoothamnium niveum and side branches that are additional cells of Z. niveum. Attached to the branches are zooids that consist each of a central cell of Z. niveum covered with ectosymbiotic bacteria. Zooids are of two types. Microzoids occur along the branches are vegetative, providing nutrients and energy to the other colony parts. Macrozooids develop at the base of each branch and are released as “swarmers” for reproductive and dispersal functions."

The above description may not be accurate, but something similar would be helpful to readers upon their first reading of the paper. By the time the reader sees Fig. 4 it becomes clearer, I suppose.

Fig. 1, in the figure the base of the stalk is black on black and does not show. Was there a last minute color reversal? (I.e.- a negative image.) To me, S1Fig is more helpful.

S Tables, how did you measure sulfide and salinity? Probably you can insert that information as a footnote to a Table.

PLOS may want to publish my reviews. I suggest, "No." These are private comments to you.

7. PLOS authors have the option to publish the peer review history of their article (what does this mean?). If published, this will include your full peer review and any attached files.

Reviewer #1: No

---

## [Author Response · Author response to Decision Letter 1]

7 Jan 2022

> Reviewer #1: It's a good revision. I recommend acceptance. However, I

> ask the authors to consider the following suggestions.

> 

> line 81. Addition of Fig. 1 has helped me to understand what the

> symbiosis looks like, but for readers who are unfamiliar with it (such

> as me) zooids are still not well described upfront. My dictionary

> defines zooids as individual units of colony, but in the present case

> the description could be more precise. That would make the paper more

> accessible. Please consider something like: "Each colony consists of a

> central stalk of the giant ciliate Zoothamnium niveum and side branches

> that are additional cells of Z. niveum. Attached to the branches are

> zooids that consist each of a central cell of Z. niveum covered with

> ectosymbiotic bacteria. Zooids are of two types. Microzoids occur along

> the branches are vegetative, providing nutrients and energy to the other

> colony parts. Macrozooids develop at the base of each branch and are

> released as "swarmers" for reproductive and dispersal functions."

> The above description may not be accurate, but something similar would

> be helpful to readers upon their first reading of the paper. By the time

> the reader sees Fig. 4 it becomes clearer, I suppose.

> S Tables, how did you measure sulfide and salinity? Probably you can

> insert that information as a footnote to a Table.

We tried to clarify and changed this paragraph to:

L 81 Zoothamnium colonies consist of a stalk and alternating branches on which individual cells grow: feeding cells called microzooids, dividing cells called terminal zooids, and macrozooids, cells responsible for asexual reproduction ([11], Fig 1). The vertical transmission of the ectosymbiont is through macrozooids. These host propagules are released as swarmers into the pelagial for dispersal. Once settled, the swarmer transforms into the terminal zooid and begins to produce the stalk and to divide, producing the terminal zooid of each branch. Nourishing microzooids are produced through division of the terminal zooid on each branch, increasing the length of the branch. Macrozooids develop at the base of the branch. These macrozooids leave the colony as swarmers as soon as a ciliary band has formed. 

> Fig. 1, in the figure the base of the stalk is black on black and does

> not show. Was there a last minute color reversal? (I.e.- a negative

> image.) To me, S1Fig is more helpful.

In figure 1 we outlined the lower part of the stalk now, which is barely visible due to the lack of the white symbionts. 

We added in the description: 

L98 Note that the lower part of the stalk, lacking white symbionts, is outlined.

---

## [Decision Letter · Decision Letter 2]

7 Feb 2022

Host-symbiont stress response to lack-of-sulfide in the giant ciliate mutualism

PONE-D-21-21486R2

Dear Dr. Espada-Hinojosa,

We’re pleased to inform you that your manuscript has been judged scientifically suitable for publication and will be formally accepted for publication once it meets all outstanding technical requirements.

Kind regards,

Marcos Pileggi, Ph.D

Academic Editor

PLOS ONE

Additional Editor Comments (optional):

Reviewers' comments:

Reviewer's Responses to Questions

**Comments to the Author**

1. If the authors have adequately addressed your comments raised in a previous round of review and you feel that this manuscript is now acceptable for publication, you may indicate that here to bypass the “Comments to the Author” section, enter your conflict of interest statement in the “Confidential to Editor” section, and submit your "Accept" recommendation.

Reviewer #1: All comments have been addressed

2. Is the manuscript technically sound, and do the data support the conclusions?

Reviewer #1: Yes

3. Has the statistical analysis been performed appropriately and rigorously? 

Reviewer #1: Yes

4. Have the authors made all data underlying the findings in their manuscript fully available?

Reviewer #1: Yes

5. Is the manuscript presented in an intelligible fashion and written in standard English?

Reviewer #1: Yes

6. Review Comments to the Author

Reviewer #1: Good! I have no further comments.

Hey, the plos one machine just rejected my review for being too short.

7. PLOS authors have the option to publish the peer review history of their article (what does this mean?). If published, this will include your full peer review and any attached files.

Reviewer #1: No

---

## [Editor Report · Acceptance letter]

16 Feb 2022

PONE-D-21-21486R2 

Host-symbiont stress response to lack-of-sulfide in the giant ciliate mutualism 

Dear Dr. Espada-Hinojosa:

I'm pleased to inform you that your manuscript has been deemed suitable for publication in PLOS ONE. Congratulations! Your manuscript is now with our production department. 

Kind regards, 

on behalf of

Dr. Marcos Pileggi 

Academic Editor

PLOS ONE